# EPA: Boosting Event-based Video Frame Interpolation with Perceptually Aligned Learning

**Yuhan Liu**[1,2], **Linghui Fu**[2], **Zhen Yang**[2], **Hao Chen**[3], **Youfu Li**[4,5], **Yongjian Deng**[2,5*]

[1]Key Laboratory of Multimedia Trusted Perception and Efficient Computing, Ministry of Education of China,
Xiamen University, 361005, P.R. China
[2]College of Computer Science, Beijing University of Technology
[3]Key Lab of Computer Network and Information Integration, Southeast University
[4]Department of Mechanical Engineering, City University of Hong Kong, Kowloon, Hong Kong SAR
[5]CityU Shenzhen Research Institute, Shenzhen, P.R. China
[1]yuhanliu@stu.xmu.edu.cn, {fulinghui@emails., yangzhen@, yjdeng@}bjut.edu.cn,
haochen303@seu.edu.cn, meyfli@cityu.edu.hk

## Abstract

Event cameras, with their capacity to provide high temporal resolution information between frames, are increasingly utilized for video frame interpolation (VFI) in challenging scenarios characterized by high-speed motion and significant occlusion. However, prevalent issues of blur and distortion within the keyframes and ground truth data used for training and inference in these demanding conditions are frequently overlooked. This oversight impedes the perceptual realism and multi-scene generalization capabilities of existing event-based VFI (E-VFI) methods when generating interpolated frames. Motivated by the observation that semantic-perceptual discrepancies between degraded and pristine images are considerably smaller than their image-level differences, we introduce EPA. This novel E-VFI framework diverges from approaches reliant on direct image-level supervision by constructing multilevel, degradation-insensitive semantic perceptual supervisory signals to enhance the perceptual realism and multi-scene generalization of the model's predictions. Specifically, EPA operates in two phases: it first employs a DINO-based perceptual extractor, a customized style adapter, and a reconstruction generator to derive multi-layered, degradation-insensitive semantic-perceptual features ($\mathcal{S}$). Second, a novel Bidirectional Event-Guided Alignment (BEGA) module utilizes deformable convolutions to align perceptual features from keyframes to ground truth with inter-frame temporal guidance extracted from event signals. By decoupling the learning process from direct image-level supervision, EPA enhances model robustness against degraded keyframes and unreliable ground truth information. Extensive experiments demonstrate that this approach yields interpolated frames more consistent with human perceptual preferences. Codes are available at https://github.com/yuhan0802/EPA.

## 1 Introduction

Recent years, Event-based Video Frame Interpolation (E-VFI) has attracted significant attention due to its outstanding performance under extreme conditions. Leveraging the high temporal resolution of event cameras [23, 48], E-VFI demonstrates clear advantages over frame-based VFI methods in challenging scenarios involving fast motion, severe occlusion, and non-rigid object deformation. It enables more accurate inter-frame motion estimation [40, 41, 21, 31], or even directly provides temporal priors at the interpolation location [28, 8], thereby facilitating the generation of higher-quality intermediate frames. However, previous works often overlook a critical issue exposed in such challenging environments: the presence of motion blur and image degradation caused by the inherent

---

* Corresponding author

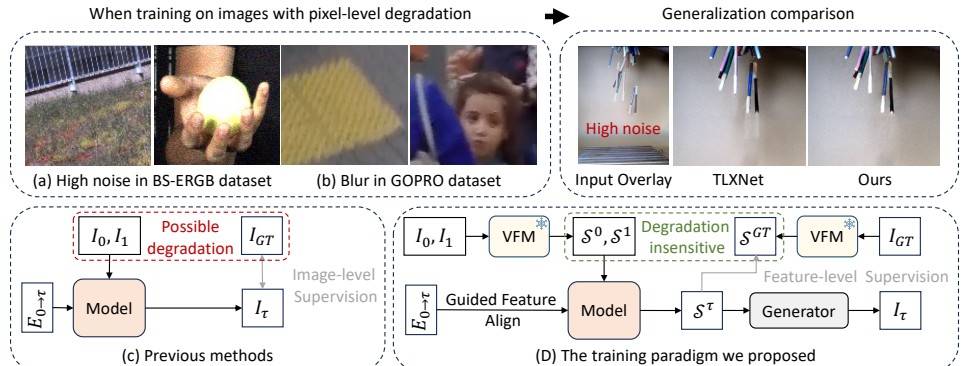

Figure 1: Visualization of image degradation in public datasets, along with a comparative illustration of our training paradigm versus prior methods, where VFM denotes the visual foundation model with frozen weights..

limitations of conventional imaging sensors, as shown in Fig. 1 (a)&(b). These methods typically assume that input frames are high quality and rely on image-level supervision during training (Fig. 1 (c)). *In fact, when images are degraded to a certain extent, defect messages conveyed by keyframes or Ground Truth data (GT) would hamper the capability of model to fit the distribution of real world, ultimately hindering the perceptual realism and generalization ability of E-VFI models across diverse scenes.*

Taking inspiration from [18] that semantic-perceptual features ($\mathcal{S}$) of images suffering less from image degradation, in this work, we propose a novel learning framework, EPA, to address above issues of **E**-VFI tasks through **P**ercepual-based feature **A**lignment. Unlike prior methods that rely on image-level supervision (Fig. 1(c)), our approach adopts feature-level training to mitigate the perceptual degradation caused by overfitting to low-quality inputs, as illustrated in Fig. 1(d). There are two stages contained in the EPA. The first stage aims to utilize the visual foundation model ($M_f$) to extract semantic-perceptual features ($\mathcal{S}^{GT}$) from GT and use the degradation-insensitive advantage of $\mathcal{S}^{GT}$ for model optimization. Here, one more thing has to be guaranteed that a carefully designed decoder can reconstruct the $\mathcal{S}^{GT}$ to its original image format. To this end, we introduce a reconstruction generator ($G_r$) equiped with a customized style adapter ($A_s$) for image reconstruction, where some low-level cues or non-saliency regions neglected by $M_f$ can be supplemented via $A_s$. The second stage is imposed for aligning the distribution gap bettween semantic-perceptual features from keyframes and Ground Truth data. We achieve this by proposing a Bidirectional Event-Guided Alignment (BEGA) module, which performs alignment process under the guidance by inter-frame temporal messages of event data in a hierachical manner. Finally, the aligned perceptual feature is fed into the generator $G_r$ for interpolating estimation.

Our contributions can be summarized as follows: (1) We propose EPA, a novel E-VFI framework that learns in the semantic-perceptual feature space using degradation-insensitive supervision, enabling more perceptually realistic frame synthesis. (2) We introduce a Style Adapter to enhance low-level details and non-salient regions overlooked by vision foundation models, improving reconstruction quality. (3) We design a Bidirectional Event-Guided Alignment (BEGA) module, which utilizes fine-grained motion cues from events to guide hierarchical feature alignment. (4) Extensive experiments on synthetic and real-world datasets demonstrate that our method consistently outperforms prior approaches in perceptual quality and generalization.

## 2 Related Work

### 2.1 Video Frame Interpolation

Motion-based frame interpolation methods dominate traditional VFI, typically relying on optical flow to warp keyframes. Enhancements such as bidirectional flow [34, 17], coarse-to-fine refinement [37], correlation-based updates [24, 35], and motion-synthesis coupling [16, 12] have improved performance, yet these methods remain vulnerable to severe occlusions. To address this, synthesis-based approaches [38, 20] avoid warping errors but demand more temporal information, increasing complexity. Both paradigms struggle under large motions due to substantial inter-frame gaps.

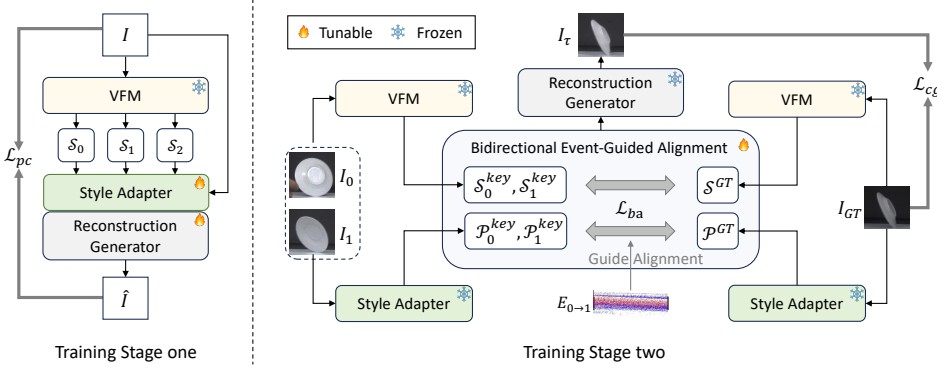

Figure 2: The pipline of the proposed EPA. Modules with the same name share weights.

Recently, generative methods [15, 45, 4] have emerged to produce perceptually convincing frames, albeit often sacrificing pixel-level accuracy due to their stochastic nature.

## 2.2 Event-based Video Frame Interpolation

Event cameras, with microsecond-level temporal resolution, enable dense inter-frame reconstruction and have driven rapid progress in event-based video frame interpolation (E-VFI). Most methods exploit events to improve optical flow estimation [40, 46, 25, 49, 51, 43, 13, 29, 27]. Advances such as spline-based modeling [41], unsupervised consistency [10], recurrent architectures [39], multi-cost volumes [21], and piecewise flow fitting [31] have improved motion estimation, yet non-rigid motion and occlusion remain challenging. To mitigate these issues, synthesis-based approaches [8, 28] have emerged, avoiding flow-related errors. However, the issue of real semantic perception deviation caused by frame degradation remains largely unaddressed. Although some methods [39] jointly optimize denoising and interpolation, this multi-task formulation tends to compromise task-specific performance and introduces unnecessary complexity.

## 3 Method

This section aims to describe the E-VFI framework, EPA, from problem formulation to detailed architectural designs. Given two input keyframes $I_0$ and $I_1$ along with the events $\mathcal{E}_{0 \to 1}$ between them, our objective is to generate the intermediate frame $I_\tau$ at a specific timestamp $\tau \in [0, 1]$. In detail, as shown in Fig. 2, EPA is composed of two stages:

The first stage contains three components, *i.e.,* a Vision Foundation Model $M_f$, a Style Adapter $A_s$, and a reconstruction generator $G_r$, where $M_f$ is for extracting semantic-perceptual features $\mathcal{S}$ from images and $A_s$ & $G_r$ are introduced for reconstruct high-fedility images from corresponding $\mathcal{S}$. In the second stage, $\mathcal{E}_{0 \to 1}$ is split into two subsets $E_{0 \to \tau}$ and $E_{1 \to \tau}$, which are then converted into voxel grids $V_{0 \to \tau}$ and $V_{1 \to \tau}$. These event voxel representations, together with the semantic features $\{\mathcal{S}_i^{key} | i \in \{0, 1\}\}$ extracted from $I_0$ and $I_1$ are fed into the Bidirectional Event-Guided Alignment (BEGA) module, which performs hierarchical alignment under the temporal guidance of event data to fit the semantics of interpolated frame. Finally, $G_r$ synthesizes the interpolated frame $I_\tau$ directly from the fitted semantic perceptual features. In the remainder of this section, we first motivate our choice of feature-level supervision in Sec. 3.1, then describe our image reconstruction strategy in Sec. 3.2, detail the design of feature alignment in Sec. 3.3, and conclude with the architectural specifics of each component in Sec. 3.4.

### 3.1 The Motivation of Introducing Feature-Level Supervision

State-of-the-art E-VFI methods [40, 41, 31, 21] typically estimate optical flow from events and warp keyframes to synthesize intermediate frames. However, optical flow mainly captures pixel-level motion, normally ignores object semantics and scene structure, making these methods heavily reliant on high-quality keyframes and GT. When keyframes suffer degradation, their discrepancy from human perception of the real world grows, and training on degraded images propagates these defects, undermining perceptual fidelity and generalization across diverse scenes. Inspired by [18], we find that the perceptual discrepancy between degraded and clean images is reduced in the feature space

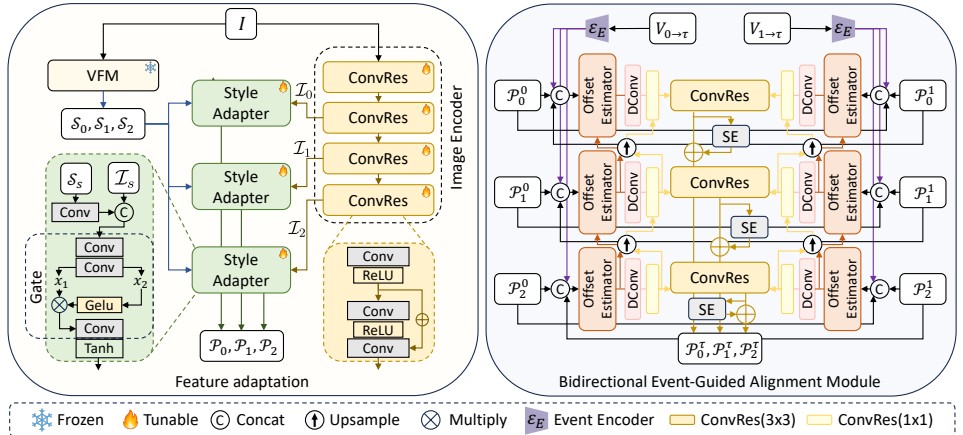

Figure 3: An overview of the proposed modules. Upsample refers to up-sampling using bilnear interpolation.

compared to the image space (Fig. 8). This motivates us proposing the feature-level supervision strategy, which reduces reliance on the quaility of input frames.

While prior works [40, 8, 28] attempt to achieve this goal by applying image-level perceptual losses [19] on the predictions, their interpolation processes still operate in the pixel domain. As a result, the models lack a true understanding of image semantics and tend to overfit to low-level details, which limits their robustness under degraded conditions. In contrast, our method enforces alignment with the ground truth at the feature level, promoting high-level semantic consistency and enhancing resilience to variations in input quality. In specific, we incorporate a vision foundation model $M_f$ that is designed to comprehensively capture object-level semantics in visual scenes [1], which has been shown to be degradation-insensitive [26]. From $M_f$, we extract semantic-perceptual features $\mathcal{S}$, which play a pivotal role in guiding the learning process.

### 3.2 Image Reconstruction from Semantic-Perceptual Features

After obtaining the semantic representation $\mathcal{S}$ from $M_f$, the network that is able to reconstruct a high-quality image from this high-level abstraction is required. To this end, we employ an off-the-shelf reconstruction generator $G_r$ that is compatible to multi-level feature input for restoring the image from $\mathcal{S}$.

**Reconstruction with Style Adapter.** From our observation, while feature representations from $M_f$ has already offer strong semantic expressiveness and generalization, they are typically optimized for high-level tasks such as segmentation and detection. Consequently, these features tend to overlook low-level cues and non-salient regions. As illustrated in the feature visualizations in Fig. 8, attention to areas like the sky is significantly reduced, which adversely affects reconstruction quality. To compensate for this deficiency, we introduce a customized style adapter $A_s$, which complements the high-level semantic features with essential low-level details. As shown in Fig. 3, we first employ a lightweight image encoder to extract low-level features $\mathcal{I}_s$, which are subsequently passed to $A_s$ to enrich the semantic representation. Specifically, the proposed style adapter $A_s$ consists of convolutional layers and a gating mechanism, which performs a weighted fusion of $\mathcal{S}_s$ and $\mathcal{I}_s$ to produce the final adapted features $\mathcal{P}_s$, where $s \in 0, 1, 2$. These adapted features enable the $G_r$ to recover high-fidelity images with both semantic consistency and fine-grained visual quality through the adapted feature $\mathcal{P}$.

### 3.3 Feature-Level Alignment with Event-Based Assistance

In the E-VFI setting, the objective is to synthesize the intermediate frame $I_\tau$ from the keyframes $I_0$ and $I_1$ along with the inter-frame events $\mathcal{E}_{0\to1}$. Unlike methods that operate at the image-level, EPA interpolates in the semantic-perceptual feature space by estimating the semantic representation of the intermediate frame and generating the final frame using a pretrained reconstruction generator $G_r$.

**Bidirectional Event-Guided Alignment (BEGA) module.** In order to obtain the semantic-perceptual features of the interpolations, we introduce the Bidirectional Event-Guided Alignment (BEGA) module. This module semantically aligns the two keyframes by leveraging the fine-grained temporal cues embedded in the event stream. Intuitively, BEGA should be directly applied to align

$\mathcal{S}^\tau$ from $\mathcal{S}_i^{key}$. However, to compensate for the fine-grained low-level semantics (*e.g.,* color and texture) that are missing in $\mathcal{S}^\tau$ during reconstruction, alignment of the $\mathcal{I}^\tau$ is also required. Our empirical findings reveal that this process is non-trivial due to the inherent limitations of event data: events are insensitive to smooth regions and lack color information. This leads to unstable training and sub-optimal reconstruction quality. Motivated by this observation, we instead apply BEGA to jointly align both semantic and low-level features, *i.e.,* direct alignment $\mathcal{P}^\tau$. This approach preserves richer low-level cues, resulting in more stable training and improved synthesis performance.

As illustrated in Fig.3, BEGA takes the event voxel grids $V_{0\to\tau}$ and $V_{1\to\tau}$ as inputs and processes them through a shared-weight event encoder $\mathcal{E}_E$ to extract hierarchical features $E_s^0$ and $E_s^1$. To retain fine motion cues inherent in the event streams, $\mathcal{E}_E$ is deliberately kept lightweight, comprising only three convolutional layers to emphasize low-level feature representation. Subsequently, at multiple spatial scales $s \in \{0, 1, 2\}$, we synthesize the intermediate frame features via one-way warping using deformable convolutions [2], followed by occlusion-aware refinement with a lightweight SE layer [11]. This design enables robust bidirectional alignment and effective fusion of intermediate semantic features, as formulated in Eq.1.

$$
\begin{aligned}
\epsilon_s^o &= \mathcal{F}_{DC}(\mathcal{F}_{OE}(\mathcal{C}(\mathcal{P}_s^o, E_s^o))), o \in \{0, 1\} \\
\epsilon_s &= \mathcal{C}(\epsilon_s^0, \epsilon_s^1), \mathcal{P}_s^\tau = \epsilon_s + \mathcal{F}_{SE}(\epsilon_s)
\end{aligned}
\tag{1}
$$

where $\mathcal{F}_{DC}$ denotes the deformable convolution, $\mathcal{F}_{OE}$ represents the offset estimator, which computes the offset and mask required by $\mathcal{F}_{DC}$, following a structure similar to that of [42]. $\mathcal{F}_{SE}$ refers to the squeeze-and-excitation operation.

## 3.4 Network Structure

Specifically, we use a DINO-pretrained ResNet backbone as the vision foundation model ($M_f$) to extract perceptual features, and adopt a normalized flow-based generator [45] as the reconstruction network ($G_r$). During the process, keyframes are fed into $M_f$ to extract multi-scale semantic features $\mathcal{S}_s^{key}$, which are enhanced by three independent style adapters $A_s$ using low-level features $\mathcal{I}_s^{key}$ from an image encoder, producing adapted features $\mathcal{P}_s$ for each scale $s \in \{0, 1, 2\}$. These are then passed to $G_r$ to synthesize the intermediate frame $I_\tau$.

## 3.5 Loss

As shown in Fig. 2, in the first stage, we utilize $\mathcal{L}_{pc}$ to ensure the reconstruction consistency, as depicted in Eq.2.

$$
\mathcal{L}_{pc} = \mathcal{L}_{Lpips}(I, \hat{I}) + \mathcal{L}_{Lap}(I, \hat{I}) + \mathcal{L}_{nll},
\tag{2}
$$

where the $\hat{I}$ denoted the the reconstructed image. The $\mathcal{L}_{Lpips}$ is perceptual loss introduced in [49], which excels at measuring the structural similarity between images. The $\mathcal{L}_{Lap}$ is a variant of the $L1$ loss proposed in [14], where the $L1$ loss is computed on the Laplacian pyramid representations of two images. $\mathcal{L}_{nll}$ is the negative log-likelihood loss, used in the optimization of normalized flow-based generators by [45].

In the second stage, we utilize a hybrid loss to emphasize feature consistency, as depicted in Eq. (3).

$$
\begin{aligned}
\mathcal{L}_{bf} &= \sum_{s=0}^{2} \mathcal{L}_2(\mathcal{P}_s, \mathcal{P}_s^{GT}) + \lambda_2 * (\mathcal{L}_2(\epsilon_0^s, \mathcal{P}_s^{GT}) + \mathcal{L}_2(\epsilon_1^s, \mathcal{P}_s^{GT})) \\
\mathcal{L}_{cg} &= \mathcal{L}_{Lpips}(I_\tau, I_{GT}) + \lambda_{lap} * \mathcal{L}_{Lap}(I_\tau, I_{GT})
\end{aligned}
\tag{3}
$$

where $\lambda_{lap}$ and $\lambda_2$ are set as 0.2, 0.1 respectively. The function $\mathcal{L}_2$ denotes the $L2$ loss that used to align two features.

Table 1: Performance comparison on synthetic datasets. The best results are marked in **Bold** while the second ones are marked with underlines. We reconstructed all skipped frames for GOPRO.

| Methods | Vimeo90k | | | GOPRO | | | | | |
| | 1 skip | | | 7 skip | | | 15 skip | | |
| | LPIPS↓ | FloLPIPS↓ | DISTS↓ | LPIPS↓ | FloLIPIS↓ | DISTS↓ | LPIPS↓ | FloLIPIS↓ | DISTS↓ |
|---|---|---|---|---|---|---|---|---|---|
| RIFE | 0.021 | 0.062 | 0.048 | 0.029 | 0.100 | 0.060 | 0.051 | 0.168 | 0.082 |
| UPR-Net | 0.015 | 0.039 | 0.037 | 0.024 | 0.077 | 0.052 | 0.042 | 0.140 | 0.067 |
| Timelens | 0.022 | 0.040 | 0.052 | 0.009 | 0.033 | 0.031 | 0.012 | 0.047 | 0.036 |
| CBMNet | 0.012 | 0.021 | 0.039 | 0.012 | 0.050 | 0.046 | 0.013 | 0.058 | 0.050 |
| TLXNet | 0.089 | 0.142 | 0.116 | 0.028 | 0.052 | 0.049 | 0.031 | 0.063 | 0.053 |
| EPA (ours) | **0.007** | **0.012** | **0.036** | **0.006** | **0.021** | **0.019** | **0.008** | **0.031** | **0.023** |

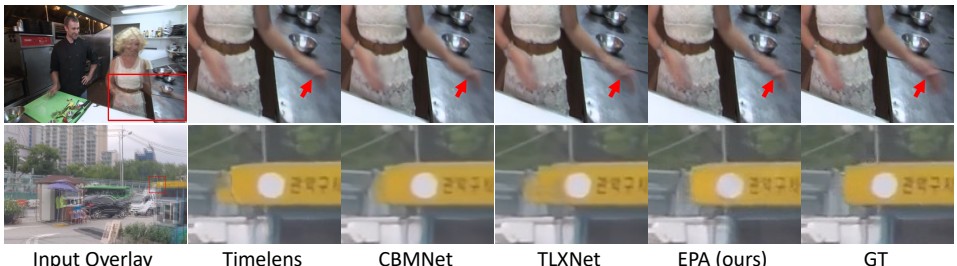

Input Overlay     Timelens     CBMNet     TLXNet     EPA (ours)     GT

Figure 4: Visual comparison among different methods on synthesis datasets.

## 4 Experiments

### 4.1 Setup

**Training Settings.** For the proposed EPA, we first optimize the style adapter and reconstruction generator modules, after which their weights are frozen to train the bidirectional feature alginement module. In the first training stage, our method is optimized using AdamW [30] for 100 epochs within the PyTorch [36]. The initial learning rate is set to $1\times10^{-4}$ and is gradually decreased to $1\times10^{-6}$ via cosine annealing. The batch size is set to 40 for each training step. In the second stage, the bidirectional adaptation module is trained for 40 epochs under the same configuration. The entire model is trained on GOPRO [33] following [31], where synthetic event data is generated using the v2e simulator [9]. Note that the training of the generator is not restricted to any specific dataset. To ensure reliable generation quality, this work applies NIQE [32] to filter out degraded images, retaining only high-quality samples. For data augmentation, both input frames and their corresponding event voxel grids are cropped to $256 \times 256$ and randomly augmented with rotation and flipping. Our normalized flow generation module follows the setup used in [45].

**State-of-the-Art Methods.** We compare our approach against several state-of-the-art VFI&E-VFI methods, including RIFE (ECCV'2022) [14], UPR-Net (CVPR'2023) [16], TimeLens (CVPR'2021) [40], CBMNet (CVPR'2023) [22], and TLXNet (ECCV'2024) [31], using their publicly available implementations. Additionally, to ensure a fair comparison, we re-train all competing methods under the same configuration. As TLXNet does not support 6-skip training, we train this method using only GOPRO. For works that have not released official code but may demonstrate promising performance [46, 28], we attempted to reproduce them. However, due to unsatisfactory results, we do not include them in our comparisons.

**Datasets.** We evaluate our method on both synthetic and real-world event datasets. The synthetic benchmarks include Vimeo90k-Triplet [47] and GOPRO [33]. For real-world evaluation, we use HS-ERGB [40], comprising 15 scenes and various motion types; BS-ERGB [41], characterized by noise and complex non-rigid deformations; and EventAid-F [7], containing various motion scenarios. *Note that due to space constraints, full evaluation results on EventAid-F are provided in the supplementary material.*

**Metrics.** Our goal is to generate images that align with human perceptual preferences, which are not fully captured by traditional image quality metrics as evidenced in Fig. 6. Thus, We primarily adopt the following metrics for performance evaluation: LPIPS [50], DISTS [6], and FloLPIPS [3], as they

Table 2: Performance comparison on real datasets. The best results are marked in **Bold** while the second ones are marked with underlines.

| Method | HS-ERGB | | | | | | | | | |
| | 5 skip | | | | | 7 skip | | | | |
| | PSNR↑ | SSIM↑ | LPIPS↓ | FloLPIPS↓ | DISTS↓ | PSNR↑ | SSIM↑ | LPIPS↓ | FloLPIPS↓ | DISTS↓ |
|---|---|---|---|---|---|---|---|---|---|---|
| RIFE | 32.624 | 0.857 | 0.032 | 0.192 | 0.083 | 31.150 | 0.836 | 0.037 | 0.212 | 0.092 |
| UPR-Net | 32.235 | 0.857 | 0.075 | 0.170 | 0.075 | 30.689 | 0.834 | 0.085 | 0.188 | 0.081 |
| Timelens | 32.760 | 0.861 | 0.046 | 0.112 | 0.059 | 31.871 | 0.851 | 0.053 | 0.126 | 0.065 |
| CBMNet | 32.206 | 0.842 | 0.098 | 0.212 | 0.108 | 31.876 | 0.837 | 0.101 | 0.218 | 0.110 |
| TLXNet | - | - | - | - | - | 31.578 | 0.827 | 0.046 | 0.105 | 0.054 |
| EPA (ours) | **33.842** | **0.872** | **0.014** | **0.057** | **0.045** | **33.402** | **0.867** | **0.015** | **0.062** | **0.048** |

| Method | BS-ERGB | | | | | | | | | |
| | 1 skip | | | | | 3 skip | | | | |
| | PSNR↑ | SSIM↑ | LPIPS↓ | FloLPIPS↓ | DISTS↓ | PSNR↑ | SSIM↑ | LPIPS↓ | FloLPIPS↓ | DISTS↓ |
|---|---|---|---|---|---|---|---|---|---|---|
| RIFE | 25.616 | 0.765 | 0.098 | 0.310 | 0.067 | 23.435 | 0.728 | 0.114 | 0.357 | 0.073 |
| UPR-Net | 25.621 | 0.779 | 0.104 | 0.308 | 0.083 | 23.081 | 0.736 | 0.108 | 0.335 | 0.082 |
| Timelens | 27.164 | 0.783 | 0.052 | 0.153 | 0.065 | 25.855 | 0.765 | 0.064 | 0.202 | 0.075 |
| CBMNet | 29.257 | **0.814** | 0.060 | 0.203 | 0.087 | 28.446 | **0.807** | 0.063 | 0.221 | 0.090 |
| TLXNet | **29.298** | 0.813 | 0.047 | 0.088 | 0.052 | **28.720** | **0.807** | 0.046 | 0.090 | 0.058 |
| EPA (ours) | 27.943 | 0.791 | **0.024** | **0.068** | **0.051** | 27.221 | 0.782 | **0.028** | **0.082** | **0.057** |

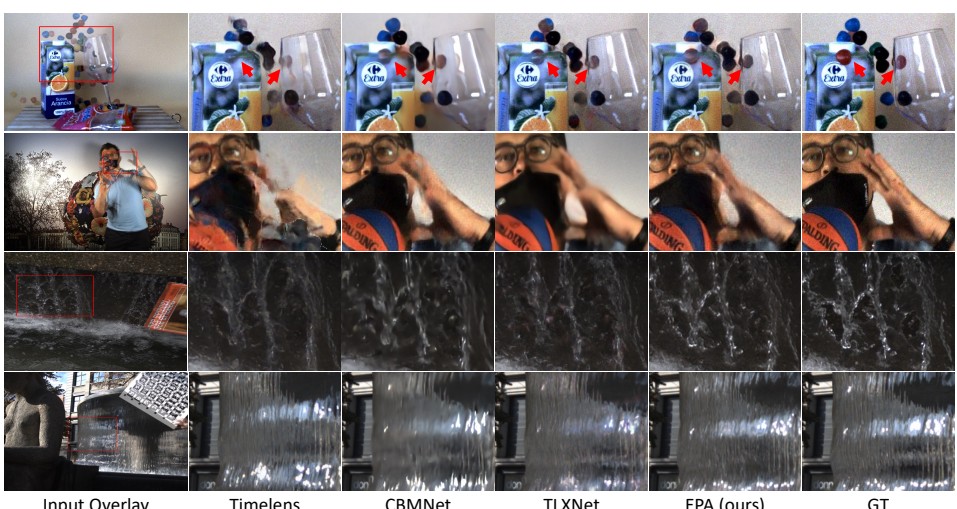

| Input Overlay | Timelens | CBMNet | TLXNet | EPA (ours) | GT |

Figure 5: Visual comparison among different methods on real datasets.

better reflect human perceptual judgments of interpolation quality. For completeness, we also include results using conventional metrics such as PSNR and SSIM [44].

## 4.2 Evaluations on Synthesis Datasets

**Quantitative Comparison** Tab. 1 presents a comparative analysis between EPA and existing methods. Thanks to our training strategy and the precise guidance from events, our method consistently achieves the highest perceptual quality on both the Vimeo90k and GOPRO datasets. Notably, on the GOPRO dataset, EPA demonstrates superior alignment between scene content and human perception. This advantage stems from EPA's semantic feature-based fitting, which enables more effective capture of scene semantics and leads to significantly better performance in terms of the DISTS metric compared to other approaches.

**Qualitative Comparison** Fig. 4 compares E-VFI methods on a synthetic dataset. Our method outperforms TimeLens, CBMNet, and TLXNet under blurred keyframes, producing clearer fingers and roofs. This superiority is attributed to our model's enhanced capability to capture scene semantics, enabling the synthesis of images that are better aligned with human perception. These results validate the effectiveness of our framework designs and perceptual feature fitting approach.

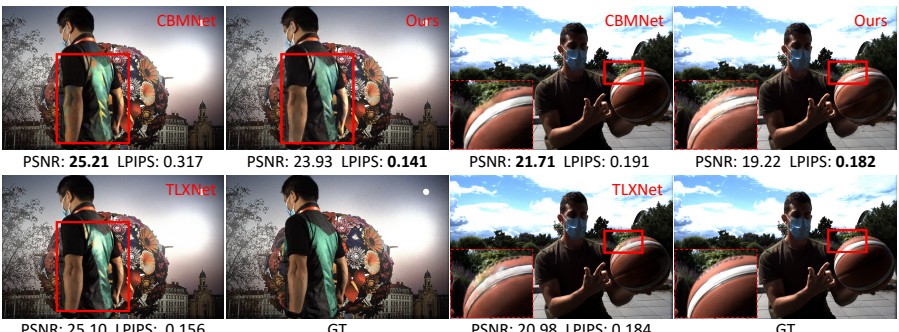

| PSNR: **25.21** LPIPS: 0.317 | PSNR: 23.93 LPIPS: **0.141** | PSNR: **21.71** LPIPS: 0.191 | PSNR: 19.22 LPIPS: **0.182** |
| PSNR: 25.10 LPIPS: 0.156 | GT | PSNR: 20.98 LPIPS: 0.184 | GT |

Figure 6: Visualization of effects and scores of inserted frames for each method in extreme motion scenes.

Table 3: Performance Comparison on EventAid-F.

| Method | Building | | Sculpture | |
| --- | --- | --- | --- | --- |
| | PSNR↑ | LPIPS↓ | PSNR↑ | LPIPS↓ |
| Timelens | **31.78** | 0.037 | **36.52** | 0.020 |
| CBMNet | 31.42 | 0.054 | 34.79 | 0.052 |
| TLXNet | 29.07 | 0.035 | 33.85 | 0.026 |
| EPA (ours) | 31.43 | **0.015** | 35.15 | **0.011** |

Figure 7: Visual comparison on EventAid-F.

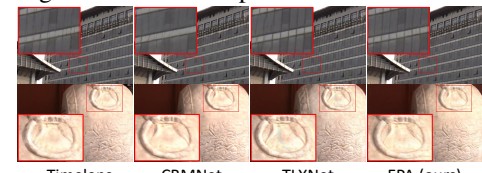

Timelens     CBMNet     TLXNet     EPA (ours)

## 4.3 Evaluations on Real Datasets

**Quantitative Comparison** Tab. 2 and Tab. 3 presents comparisons on real-world datasets, highlighting the practical effectiveness of our method. Our approach consistently leads in perceptual metrics across three datasets. Notably, on the HS-ERGB dataset, EPA also surpasses other methods in traditional image quality metrics such as PSNR and SSIM. We attribute this to the limitations of compared methods in handling the widespread irregular object deformations present in the data, while our method, leveraging semantic-level features, addresses these challenges more effectively.

**Qualitative Comparison** Fig. 5 and Fig. 7 presents a visual comparison of different E-VFI methods on a real-world event dataset. A direct comparison with the outputs of TimeLens, CBMNet, and TLXNet reveals the advantages of our method. Specifically, in Fig. 5, our approach ensures better visual quality in various extreme motion scenarios, as demonstrated by more complete candies, fingers, and better-preserved water flow details. From the images produced by CBMNet, we observe that when meet noisy keyframes or occlusion situations, the generated interpolated frames exhibit sever blur (*e.g.*, the candy) and poor-quality (*e.g.*, the water flow). Instead, our method can handle such scenerios more effectively on dynamic objects. Similarly, Fig. 7 shows that our method generates sharper and more perceptually aligned results for structures such as building and sculptures. We attribute to that benefiting from the perceptual features alignment with visual foundation models, our approach is able to synthesize more realistic images by comprehending the semantics of interpolated scenes to a certain extent. Interestingly, as illustrated in Fig. 6, although CBMNet and TLXNet achieve higher PSNR scores, our method still produces visually superior results in extreme scenarios, demonstrating the efficacy of using perceptual metrics (*e.g.*, lpips/dists) to measure the VFI task.

Table 4: Comparison results of different feature extraction settings in training stage one.

| Variants | Norm. | $A_s$ | Vimeo90k | | | |
| --- | --- | --- | --- | --- | --- | --- |
| | | | PSNR | SSIM | LPIPS | DISTS |
| A | | | 36.121 | 0.983 | 0.0038 | 0.0218 |
| B | ✓ | | 40.793 | 0.991 | 0.0016 | 0.0086 |
| C (ours) | ✓ | ✓ | **47.732** | **0.996** | **0.0002** | **0.0012** |

Table 5: Comparison results of different $\mathcal{S}^\tau$ synthesis method settings in training stage two. $\mathcal{E}$ denotes events.

| Variants | $\mathcal{E}$ | SCA | BEGA | Vimeo90k | | | | |
| --- | --- | --- | --- | --- | --- | --- | --- | --- |
| | | | | PSNR | SSIM | LPIPS | FloLPIPS | DISTS |
| D | | | | 28.834 | 0.857 | 0.0215 | 0.0481 | 0.0755 |
| E | ✓ | ✓ | | 36.900 | 0.961 | 0.0078 | 0.0129 | 0.0375 |
| F (ours) | ✓ | | ✓ | **37.011** | **0.964** | **0.0074** | **0.0124** | **0.0361** |

## 5 Model Analysis

In this section, we conduct experiments on the Vimeo90k dataset to analyze the effectiveness of the two core components in our proposed method: the Style Adapter and the Bidirectional Event-Guided Alignment module.

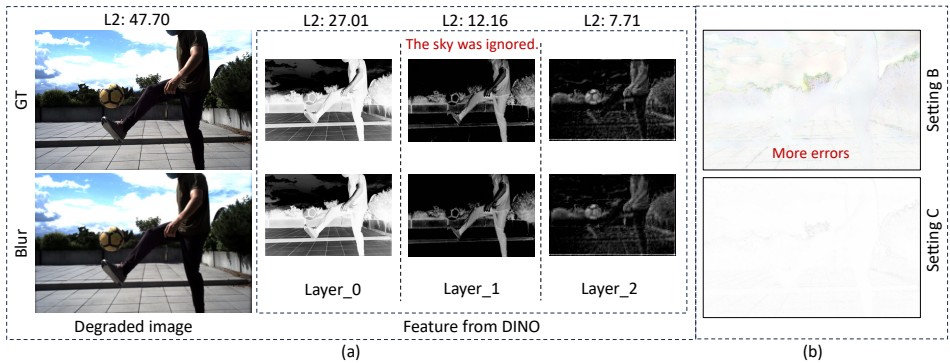

Figure 8: Visualization of comparison results. (a) compares L2 distances between image-level and feature-level supervision, alongside DINO feature visualizations. (b) shows difference maps between the generated results from settings B and C in Tab. 4 and the ground truth.

**Rationality and effectiveness of feature-level supervision.** In Fig. 8(a), we present a comparison between the differences induced by image-level supervision and those induced by feature-level supervision when an image undergoes image-level degradation, exemplified here by blurring. We use the L2 distance as an evaluation metric. It can be observed that the feature-level supervision signals extracted from DINO exhibit higher robustness to blur degradation. Furthermore, as the network depth increases, the robustness of the feature supervision becomes more pronounced. This improvement is attributed to the VFM's capability to capture image representations from a perceptual perspective, which helps mitigate the impact of image-level distortions, suggesting the reasonableness of our design philosophy to a certain extent.

**Effectiveness of the Style Adapter** ($A_s$). As presented in Tab. 4, to evaluate the proposed $A_s$, we introduce baseline model A, which utilizes the first three neural blocks of features from DINO, without incorporating shallow image details. Given that DINO is pretrained on ImageNet [5] with associated normalization, we introduce model B, which applies both normalization and denormalization processes. Model B normalizes input images using ImageNet statistics and denormalizes the outputs accordingly. Model C is built upon Model B and trained in combination with the proposed $A_s$. The results of settings A and B in Tab. 4 highlight the importance of the adaptation process. Furthermore, the comparison between Settings B and C demonstrates that our choice substantially improves the model's ability in reconstructing images from perceptual features, thereby enhancing the quality of the synthesized interpolated frames, as illustrated in Fig. 8(b).

**Effectiveness of the Bidirectional Event-Guided Alignment (BEGA) Module.** As shown in Tab. 5, to verify the importance of event guidance, we construct a baseline model D by removing all event data and performing feature fitting purely based on frames. Additionally, to assess the effectiveness of our specific design, we introduce a comparison model E, which employs a fusion module—referred to as SCA—comprising a self-attention layer followed by a cross-attention layer, similar to the structure proposed in [21]. The comparison between models D and E highlights the significance of event guidance. The precise inter-frame motion priors provided by events significantly enhance the model's ability to fit features of the interpolated frame. Furthermore, the performance gap between models E and F demonstrates the superiority of our proposed module.

# 6 Conclusion

In this work, we present EPA, a novel E-VFI framework that addresses the limitations of image-level degradation in modeling real-world data distributions. By leveraging degradation-insensitive semantic supervision, EPA enhances both the perceptual fidelity and generalization capability of frame synthesis under challenging conditions. EPA's two-stage training paradigm first extracts robust semantic-perceptual features and ensures reconstruction quality via a dedicated style adapter. In the second stage, the proposed Bidirectional Event-Guided Alignment module effectively aligns semantic features from keyframes to the ground truth, guided by fine-grained event cues, and synthesizes the interpolation via a pretrained reconstruction generator. Extensive experiments demonstrate that our method outperforms existing approaches in both perceptual quality and generalization, offering a more reliable solution for real-world scenarios affected by camera noise and motion blur.

# 7 Acknowledgments

This work is jointly supported by Beijing Natural Science Foundation (4252026), National Natural Science Foundation of China (62203024, 62573369), Research and Development Program of Beijing Municipal Education Commission (KM202310005027).

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
