# OpenReview forum: "EPA: Boosting Event-based Video Frame Interpolation with Perceptually Aligned Learning"
_NeurIPS.cc/2025/Conference — NeurIPS 2025 poster_

### Official Review · Reviewer_QbtM · 2025-06-25

**Clarity:** 3
**Significance:** 2
**Originality:** 3
**Rating:** 4
**Confidence:** 5

**Summary:**

This paper addresses video frame interpolation (VFI) using event signals, highlighting that existing event-based methods often overlook blur and distortion in keyframes and ground-truth data, leading to suboptimal performance. To mitigate this, the authors propose EPA, a novel framework that leverages multilevel, degradation-insensitive semantic-perceptual supervision using a DINO-pretrained vision backbone. The approach aims to enhance perceptual realism and experiments show that EPA achieves state-of-the-art results in perceptual metrics such as LPIPS.

**Questions:**

Why is there no computational cost analysis?

How does the model perform in terms of video-level consistency?

How does it compare with recent large-motion-capable VFI methods [1,2]?

(Refer to the “Weaknesses” section)

**Ethical Concerns:**

["NO or VERY MINOR ethics concerns only"]

**Final Justification:**

Given the promise of discussion and comparisons in the main paper, as well as the computational costs and currently unavailable qualitative comparisons, this reviewer will withhold the rating.

**Limitations:**

A clear analysis of computational costs and more comprehensive video-based comparisons would strengthen the paper’s claims.

**Quality:**

3

**Strengths And Weaknesses:**

Strengths:

- The paper is well-organized and clearly written.

- The model design is technically sound.

- Using pretrained vision features for perceptual-level supervision is a reasonable.

Weaknesses:

- No analysis of computational cost is provided (e.g., model size, speed, training overhead).

- No video results are shown, which limits the evaluation of temporal consistency—a key factor in VFI tasks.

- Recent methods capable of handling large motion and ambiguous supervision, such as classic models with novel indexing [1] and diffusion-based approaches [2], are not compared. The examples shown seem to involve relatively moderate motion; a qualitative comparison on more challenging sequences would help justify the use of event signals.

- The related work section lacks coverage of prior efforts on unsupervised event-based VFI [3] for degraded inputs, and those further exploiting the hidden temporal information in degraded inputs [4–6].

Minor Issues:

There are typos such as line 111: “losse” → “losses”

References:

[1] Clearer frames, anytime: Resolving velocity ambiguity in video frame interpolation, ECCV 2024

[2] Framer: Interactive frame interpolation, ICLR 2025

[3] Unsupervised Event-Based Video Reconstruction, WACV 2024

[4] Event-Based Frame Interpolation with Ad-hoc Deblurring, CVPR 2023

[5] Unifying Motion Deblurring and Frame Interpolation with Events, CVPR 2022

[6] Event-Guided Frame Interpolation and Dynamic Range Expansion of Single Rolling Shutter Image, ACM-MM 2023

---

> ### Author Rebuttal · Authors · 2025-07-30
>
> We thank Reviewer QbtM for devoting time to this review and providing valuable comments.
>
> > ***Q1:**"Why is there no computational cost analysis?"*
>
> Thank you for the reminder. This part will be included in the supplementary material.
>
> - Training stage: The training process is composed of two distinct stages, conducted on a single NVIDIA RTX 3090:
>
>   1. Stage 1: The model was trained for 100 epochs using a single RGB image with a batch size of 28 and image size of 256x256, requiring approximately 24 hours.
>
>   2. Stage 2: The weights of the other parts were frozen, and only the BEGA module was trained for 40 epochs using two RGB images and the corresponding event streams, with a batch size of 32. This stage took about 48 hours to complete.
>
> - Inference Cost and Comparison: The table below details the resource consumption compared to other methods. Our approach achieves a highly competitive runtime and model size, demonstrating an excellent balance between performance and efficiency.
>
>
> | Method         | Parameters$\dagger$ | Flops  | Runtime | PSNR$\uparrow$ | LPIPS$\downarrow$ |
> | -------------- | ------------------ | ------ | ------- | -------------- | ----------------- |
> | Timelens       | 79.2M              | 18.7B  | 0.033s  | 31.871         | 0.053             |
> | CBMNet         | 22.2M              | 46.7B  | 0.231s  | 31.876         | 0.101             |
> | TLXNet         | 7.87M              | 16.5B | 0.027s  | 31.578         | 0.046             |
> | **EPA (ours)** | 13.9M              | 20.2B  | 0.086s  | 33.402         | 0.015             |
>
> $\dagger$ represents trainable parameters. PSNR and LPIPS scores come from the real dataset HS-ERGB.
>
> > ***Q2:**"How does the model perform in terms of video-level consistency?"*
>
> Thank you for the question.
>
> - Due to conference policy, we cannot share external links at this stage, but they will be included in the supplementary material. For a quantitative measure, we rely on the FloLPIPS metric, which is specifically designed to measure perceptual coherence across frames. As shown in Tab 1 & 2 of our paper, our method achieves state-of-the-art FloLPIPS scores on all datasets, quantitatively confirming its excellent temporal consistency.
>
>
>
> > ***Q3:**"How does it compare with recent large-motion-capable VFI methods [1,2]?"*
>
> Thank you for bringing these highly relevant, recent works to our attention. We have conducted experiments on the BS-ERGB dataset, with results shown in the table below.
>
> | Method | PSNR$\uparrow$ | LPIPS$\downarrow$ | FloLPIPS$\downarrow$ | DISTS$\downarrow$ |
> | ------ | -------------- | ----------------- | -------------------- | ----------------- |
> | [1]    | 23.240         | 0.102             | 0.312                | 0.066             |
> | [2]    | 19.262         | 0.331             | 0.457                | 0.182             |
> | Ours   | **27.221**     | **0.028**         | **0.082**            | **0.057**         |
>
>
>
> The results demonstrate our method's superior performance on this challenging real-world dataset with degradation. It is noteworthy that [1] also achieves competitive LPIPS and DISTS scores, reinforcing the importance of evaluating perceptual quality—a direction we also champion in our work. We believe this new comparison provides valuable context, and we will incorporate these important results and this analysis into the main paper.
>
>
>
> > ***Q4:**"The related work section lacks coverage of prior efforts on unsupervised event-based VFI [3] for degraded inputs, and those further exploiting the hidden temporal information in degraded inputs [4–6]."*
>
> We appreciate the reminder and agree that these citations are important for a complete literature review. We will revise the related work section in the final manuscript to include and properly discuss these papers. Thank you for helping us strengthen our paper.

---

> ### Comment · Reviewer_QbtM · 2025-08-05
>
> Thanks to the authors for their efforts.
>
> Given the promise of discussion and comparisons in the main paper, as well as the computational costs and currently unavailable qualitative comparisons, this reviewer will withhold the rating.

---

> > ### Author Response · Authors · 2025-08-06
> >
> > Thank you, Reviewer QbtM, for your detailed review and for acknowledging the promises made in our rebuttal.
> >
> > We assure you that all promised additions will be fully incorporated into the main text and supplementary materials. We believe these updates will address your concerns and significantly strengthen the paper.
> >
> > We appreciate your time and constructive feedback, and we look forward to presenting the improved revision.

---

### Official Review · Reviewer_GPzr · 2025-07-01

**Clarity:** 2
**Significance:** 3
**Originality:** 2
**Rating:** 4
**Confidence:** 4

**Summary:**

This paper proposes an event-based method for arbitrary-time video frame interpolation between two degraded images. Compared with existing flow-based or multi-task (restoration + interpolation) methods, the proposed framework claims to avoid such trade-offs without introducing additional complexity.

**Questions:**

How is the off-the-shelf reconstruction generator [41] aligned with features extracted by M_f (DINO), especially to support the claim of compatible to multi-level feature input for restoring the image?

Inconsistently with HS-ERGB, the proposed method achieves better perceptual metrics (e.g., LPIPS) but worse traditional ones (e.g., PSNR) on BS-ERGB and EventAid datasets. An in-depth analysis for this phenomenon is needed, including more significant visual comparison analyses.

The paper should clarify differences between training and inference stages and report the inference cost.

Does the claim of L82 apply to all existing methods? For example, does EVDI [46] have this problem?

minor:

111 losse

supp 46 Limition

**Ethical Concerns:**

["NO or VERY MINOR ethics concerns only"]

**Final Justification:**

I find most of my concerns addressed, except that I still believe the uniqueness of events in the framework design is unconvincing. Iwill raise my rating to borderline accept.

**Limitations:**

yes

**Quality:**

2

**Strengths And Weaknesses:**

Strengths

The paper presents a new pipeline to learn robust features from vision foundation models and guide reconstruction under image degradation.

The proposed method is extensively evaluated on both synthetic and real datasets, demonstrating superior performance over existing approaches.

Weaknesses

The proposed framework lacks a natural connection between motion-induced image degradation (e.g., blur) and event generation, and the two core modules are relatively loosely coupled.

The DINO-based feature extraction module (Sec 3.1) is primarily used to learn degradation-robust features from images and is unrelated to events. Similar strategies have been adopted in prior work. The authors should clarify the unique contribution of this component in their pipeline.

Unlike prior event-based optical flow methods where events contribute to both motion estimation and image synthesis, the proposed approach uses events only for alignment, without directly enhancing semantic features. The rationale for this design should be discussed.

The ablation study is limited to toggling modules on or off. A deeper analysis is needed, such as replacing DINO with other backbone models or combining other backbones with the proposed supervision to verify the generality of the approach.

---

> ### Author Rebuttal · Authors · 2025-07-31
>
> We thank Reviewer GPzr for devoting time to this review and providing valuable comments.
>
> > ***Q1:**"The proposed framework lacks a natural connection between motion-induced image degradation (e.g., blur) and event generation, and the two core modules are relatively loosely coupled."*
>
> Thank you for this question.
>
> - We believe the two core modules are, in fact, tightly and logically coupled to solve the problem. Our approach decomposes the complex task of "interpolating between degraded frames" into two distinct, manageable sub-problems: (1) Extracting robust semantic information from the (potentially degraded) keyframes.  (2) Accurately aligning this semantic information through reliable spatial-temporal information to generate the intermediate frame.
>
> - The key point is that only with events assistance, the model can obtain the precise inter-frame motion information required to perform the alignment in the second sub-problem. As we argue in the paper, event data's sensitivity to moving objects aligns perfectly with the changing-sensitive nature of semantic features from models like DINO. This enables a robust, temporally coherent alignment in the feature space, which is far more reliable than aligning degraded pixels. Therefore, the DINO features and the event-guided alignment are inseparable components for generating the final, high-quality interpolated frame.
>
>
>
> > ***Q2:**"The DINO-based feature extraction module (Sec 3.1) is primarily used to learn degradation-robust features from images and is unrelated to events. Similar strategies have been adopted in prior work. The authors should clarify the unique contribution of this component in their pipeline."*
>
> - The unique contribution is not VFM (e.g., DINO) itself, but how we leverage it within a new E-VFI paradigm. Previous works optimize their model predominantly rely on pixel-level supervision from the ground truth at the output image-level stage. When keyframes and the GT are degraded, this approach forces the model to learn to reproduce these degradations, harming its generalization.
>
> - Our paradigm addresses this by using VFM to lift the problem into a more robust feature space. The role of VFM (such as DINO) is to act as a stable, degradation-insensitive "perceptual anchor". The main contribution of our work is to design a novel framework that operates effectively for dealing with such inspiration, especially with event data assistance. We intentionally keep the VFM with minimal modifications, ensuring that our framework can be easily extended to other domains or tasks by simply swapping in a different, more suitable feature extractor.
>
>
>
> > ***Q3:**"Unlike prior event-based optical flow methods where events contribute to both motion estimation and image synthesis, the proposed approach uses events only for alignment, without directly enhancing semantic features. The rationale for this design should be discussed."*
>
> - This is a deliberate design choice. When keyframe degradation is the primary problem, traditional optical flow methods tend to propagate the degradation from the keyframe to the warped, interpolated frame. This results in an output that, while perhaps spatially aligned, is still perceptually poor in semantic expression. By moving to a feature-level alignment operation, we bypass this issue. We use events for what they excel at—providing precise motion cues for alignment— for facilitating the final synthesis, resulting in high-quality, degradation-insensitive outputs.
>
>
>
> > ***Q4:**"The ablation study is limited to toggling modules on or off. A deeper analysis is needed, such as replacing DINO with other backbone models or combining other backbones with the proposed supervision to verify the generality of the approach."*
>
> - To address this and further verify the generality of our approach, we have conducted a new experiment where we replace the DINO backbone with MoCo v3, another popular self-supervised vision model. After training, the model with MoCo v3 achieved a PSNR of 35.11 and an LPIPS of 0.0079 on the Vimeo90k dataset. While slightly lower than our DINO-based model, these strong results demonstrate that our proposed framework is not tied to a specific backbone and can successfully operate with different feature extractors, confirming the generality of our approach.
>
>
>
> > ***Q5:**"How is the off-the-shelf reconstruction generator [41] aligned with features extracted by M_f (DINO), especially to support the claim of compatible to multi-level feature input for restoring the image?"*
>
> - This compatibility is achieved through our two-stage training process and the data-driven nature of the network. In Stage 1, we specifically train the reconstruction generator (G_r) and the Style Adapter (A_s) to do one thing: reconstruct a high-quality image from the multi-level features provided by DINO. An MLP is used to unify the feature dimensions. This pre-training ensures that the generator learns the exact mapping from the DINO feature distribution to the image distribution. It becomes an "expert" at translating DINO's language into a visual output, ensuring the two are perfectly aligned for the main task in Stage 2.
>
>
>
> > ***Q6:**"Inconsistently with HS-ERGB, the proposed method achieves better perceptual metrics (e.g., LPIPS) but worse traditional ones (e.g., PSNR) on BS-ERGB and EventAid datasets. An in-depth analysis for this phenomenon is needed, including more significant visual comparison analyses."*
>
> Thank you for the comment.
>
> - Our goal is to generate results faithful to the underlying authentic scene, rather than the degraded ground truth.
>
> - The issue you raised is, in fact, a reflection of the success of our method rather than a flaw. The BS-ERGB dataset is well-known for its substantial noise and non-rigid deformations, and both its input frames and ground-truth references are inherently low in quality. The EventAid dataset also contains a significant amount of blur. In such conditions, a robust model should aim to produce results that are visually clean, perceptually aligned with human judgment, rather than replicating the noise and artifacts present in the original data. Traditional pixel-level metrics like PSNR tend to penalize such ‘reasonable deviations’ since they measure absolute pixel differences without considering perceptual quality.
>
> - Our method achieves significant improvements on perceptual metrics such as LPIPS and DISTS, and the qualitative results (Fig.5, Fig.6 and Fig.7) in the paper clearly show that our interpolated frames exhibit much better visual realism than those generated by prior methods. This strongly supports the claim that our model better captures ‘perceptual authenticity’ and is more suited for real-world deployment.
>
> - Moreover, our approach does not suffer from a noticeable disadvantage on traditional metrics like PSNR and SSIM, and in fact achieves competitive results. This further demonstrates the comprehensive effectiveness of our framework. Due to conference requirements, more significant visual comparison analyses will be included in the supplementary material.
>
> > ***Q7:**"The paper should clarify differences between training and inference stages and report the inference cost."*
>
> Thank you for the comment. This part will be included in the supplementary material.
>
> - Training stage: The training process is composed of two distinct stages, conducted on a single NVIDIA RTX 3090:
>
>   1. Stage 1: The model was trained for 100 epochs using a single RGB image with a batch size of 28 and image size of 256x256, requiring approximately 24 hours.
>
>   2. Stage 2: The weights of the other parts were frozen, and only the BEGA module was trained for 40 epochs using two RGB images and the corresponding event streams, with a batch size of 32. This stage took about 48 hours to complete.
>
> - Inference Cost and Comparison: In the inference stage, the entire network is used to generate the final frame. The table below details the resource consumption compared to other methods. Our approach achieves a highly competitive runtime and model size, demonstrating an excellent balance between performance and efficiency.
>
> | Method         | Parameters$\dagger$ | Flops  | Runtime | PSNR$\uparrow$ | LPIPS$\downarrow$ |
> | -------------- | ------------------ | ------ | ------- | -------------- | ----------------- |
> | Timelens       | 79.2M              | 18.7B  | 0.033s  | 31.871         | 0.053             |
> | CBMNet         | 22.2M              | 46.7B  | 0.231s  | 31.876         | 0.101             |
> | TLXNet         | 7.87M              | 16.5B | 0.027s  | 31.578         | 0.046             |
> | **EPA (ours)** | 13.9M              | 20.2B  | 0.086s  | 33.402         | 0.015             |
>
> $\dagger$ represents trainable parameters. PSNR and LPIPS scores come from the real dataset HS-ERGB.
>
> > ***Q8:**"Does the claim of L82 apply to all existing methods? For example, does EVDI [46] have this problem?"*
>
> Thank you for the comment.
>
> - The denoising and deblurring methods mentioned in [35][46], which are currently leading in the field, do exhibit similar issues. They share a similar objective with ours, aiming to alleviate degradation in key frames for E-VFI. However, they take a different approach, opting for a 'denoise-then-interpolate' strategy. While this approach can yield high-quality interpolated frames and clear key frames, the multi-stage process may introduce error accumulation and additional computational complexity, which contributes to their relatively lower performance in E-VFI tasks.
> - In contrast, our EPA framework is a single, end-to-end solution for the interpolation task. By operating entirely in a degradation-robust feature space, we circumvent the need for an explicit, separate denoising step. This avoids the potential pitfalls of error accumulation inherent in multi-stage pipelines, offering a more direct solution to the problem.

---

> ### Comment · Reviewer_GPzr · 2025-08-05
>
> Thanks for the rebuttal. I find most of my concerns addressed, except that I still believe the uniqueness of events in the framework design is unconvincing. I will raise my rating to borderline accept.

---

> > ### Author Response · Authors · 2025-08-06
> >
> > Thank you, Reviewer GPzr. We sincerely appreciate you reconsidering our work and raising our rating. We are grateful for your constructive feedback throughout this process.
> >
> > We would also like to take this final opportunity to briefly clarify why events are uniquely essential to our framework's design.
> >
> > While traditional methods use events to estimate optical flow for warping pixels, our framework introduces a new challenge: aligning abstract, semantic features. These deep features are sensitive to object semantics but lack the precise, low-level texture information that flow algorithms rely on.
> >
> > This is where events provide a unique and indispensable bridge. Their ability to precisely capture inter-frame motion trajectories at a high temporal resolution offers the exact guidance needed to robustly align these semantic feature maps through time. This task—aligning deep features using event data—cannot be effectively solved by traditional flow-based or frame-based methods, which makes events a unique and crucial component of our feature-space interpolation paradigm.
> >
> > Thank you once again for your time and valuable engagement.

---

### Official Review · Reviewer_pM6N · 2025-07-02

**Clarity:** 3
**Significance:** 3
**Originality:** 3
**Rating:** 4
**Confidence:** 4

**Summary:**

This paper proposes a novel event-based video frame interpolation framework, designed to address image distortions in complex scenarios, where existing methods often fail. By introducing multi-level, degradation-insensitive, semantic-aware supervision signals, the proposed method enhances the perceptual realism and cross-scene generalization of the interpolated frames. Experimental results demonstrate that the proposed method excels in both perceptual quality and generalization capability.

**Questions:**

See Weakness for details.

**Ethical Concerns:**

["NO or VERY MINOR ethics concerns only"]

**Final Justification:**

Thank you for your feedback, which has resolved many of my concerns. The computational efficiency and broad applicability of the proposed method need improvement in future work. My final recommendation score is Borderline accept.

**Limitations:**

Yes

**Quality:**

3

**Strengths And Weaknesses:**

Strengths:

(1) The paper proposes an effective feature-level supervision strategy that addresses the limitations of existing methods when dealing with degraded images.

(2) The experimental results not only show excellent performance on traditional metrics (e.g., PSNR, SSIM) but also significantly outperform prior works on perceptual quality metrics (e.g., LPIPS, DISTS).

Weaknesses:

(1) The proposed method employs a generator-based architecture to produce the final image, a technique inherently known to boost the perceptual quality of generated outputs. Consequently, it is unclear whether the final improvement in perceptual quality stems more from the generator architecture itself or from the proposed feature-level supervision. The authors should provide more extensive ablation studies or analysis to disentangle these two effects.

(2) The paper initially discusses "motion blur and image degradation" in broad terms, but the subsequent experiments primarily demonstrate effectiveness on blur and camera noise. Image degradation encompasses a much wider range of phenomena, such as lens distortion, rolling shutter artifacts, compression artifacts, low resolution, rain, and fog. It is questionable whether the proposed method maintains its robustness against these other types of degradation.

(3) The proposed method requires a two-stage training process, but its training efficiency is not discussed. Furthermore, the computational efficiency of the method during inference remains unclear, and it is not specified whether it can meet real-time requirements.

(4) The robustness of DINO features is not guaranteed in specialized domains, such as medical microscopy videos or materials analysis. Since the proposed method relies heavily on DINO features for semantic alignment, its broad applicability and performance in such out-of-distribution scenarios are uncertain.

---

> ### Author Rebuttal · Authors · 2025-07-30
>
> We thank Reviewer pM6N for devoting time to this review and providing valuable comments.
>
>
> > ***Q1:**"The proposed method employs a generator-based architecture to produce the final image, a technique inherently known to boost the perceptual quality of generated outputs. Consequently, it is unclear whether the final improvement in perceptual quality stems more from the generator architecture itself or from the proposed feature-level supervision. The authors should provide more extensive ablation studies or analysis to disentangle these two effects."*
>
> Thank you for your comment.
>
> - All components of the training framework we propose are interchangeable. This component primarily serves to synthesize interpolated frames, which is not our main focus. During experiments, we also evaluated U-Net-based [1] and Transformer-based [2] decoders. The U-Net structure increased the number of parameters, while the Transformer-based decoder introduced longer inference latency. Both alternatives yielded similar interpolation quality ($\pm$3%), with larger models offering marginal gains at the cost of efficiency. To strike a balance between performance and runtime, we ultimately adopted a simple yet effective generator.
>
>
> [1] Chan K C K, Zhou S, Xu X, et al. Basicvsr++: Improving video super-resolution with enhanced propagation and alignment[C]//Proceedings of the IEEE/CVF conference on computer vision and pattern recognition. 2022: 5972-5981.
>
> [2] Kim T, Chae Y, Jang H K, et al. Event-based video frame interpolation with cross-modal asymmetric bidirectional motion fields[C]//Proceedings of the IEEE/CVF Conference on Computer Vision and Pattern Recognition. 2023: 18032-18042.
>
>
>
> > ***Q2:**"The paper initially discusses "motion blur and image degradation" in broad terms, but the subsequent experiments primarily demonstrate effectiveness on blur and camera noise. Image degradation encompasses a much wider range of phenomena, such as lens distortion, rolling shutter artifacts, compression artifacts, low resolution, rain, and fog. It is questionable whether the proposed method maintains its robustness against these other types of degradation."*
>
> Thank you for the comment.
>
> - We have added comparative results for different methods under JPEG compression artifacts (Table 1) and low-resolution degradation (Table 2) in the BS-ERGB pen_03 scene. As shown, our method ensures the best perceptual quality and highly competitive traditional metrics. Although CBMNet is slightly superior in traditional metrics, it suffers a collapse in perceptual quality, with significant color errors in moving objects. TLXNet, being purely based on optical flow, preserves the degradation from key frames to the fullest, resulting in large discrepancies with the non-degraded ground truth and severely affecting its performance. This further supports our viewpoint and validates the effectiveness of our method. Due to conference requirements, the detailed comparison images will be included in the supplementary material.
>
>
>
> [Table 1]  JPEG compression artifacts:
>
> | Method   | PSNR$\uparrow$ | LPIPS$\downarrow$ | FloLPIPS$\downarrow$ | DISTS$\downarrow$ |
> | -------- | -------------- | ----------------- | -------------------- | ----------------- |
> | Timelens | 21.373         | 0.216             | 0.313                | 0.235             |
> | CBMNet   | **22.082**     | 0.575             | 0.818                | 0.277             |
> | TLXNet   | 20.126         | 0.371             | 0.599                | 0.268             |
> | Ours     | 21.423         | **0.141**         | **0.246**            | **0.177**         |
>
>
>
> [Table 2] Low resolution:
>
> | Method   | PSNR$\uparrow$ | LPIPS$\downarrow$ | FloLPIPS$\downarrow$ | DISTS$\downarrow$ |
> | -------- | -------------- | ----------------- | -------------------- | ----------------- |
> | Timelens | 21.285         | 0.191             | 0.253                | 0.247             |
> | CBMNet   | **22.022**     | 0.551             | 0.743                | 0.274             |
> | TLXNet   | 20.175         | 0.226             | 0.382                | 0.247             |
> | Ours     | 21.584         | **0.135**         | **0.212**            | **0.180**         |
>
>
>
> > ***Q3:**"The proposed method requires a two-stage training process, but its training efficiency is not discussed. Furthermore, the computational efficiency of the method during inference remains unclear, and it is not specified whether it can meet real-time requirements."*
>
> Thank you for the question. This part will be included in the supplementary material.
>
> - Training stage: The training process is composed of two distinct stages, conducted on a single NVIDIA RTX 3090:
>
>   1. Stage 1: The model was trained for 100 epochs using a single RGB image with a batch size of 28 and image size of 256x256, requiring approximately 24 hours.
>
>   2. Stage 2: The weights of the other parts were frozen, and only the BEGA module was trained for 40 epochs using two RGB images and the corresponding event streams, with a batch size of 32. This stage took about 48 hours to complete.
>
> - Inference Cost and Comparison: The table below details the resource consumption compared to other methods. Our approach achieves a highly competitive runtime and model size, demonstrating an excellent balance between performance and efficiency.
>
>
> | Method         | Parameters$\dagger$ | Flops  | Runtime | PSNR$\uparrow$ | LPIPS$\downarrow$ |
> | -------------- | ------------------ | ------ | ------- | -------------- | ----------------- |
> | Timelens       | 79.2M              | 18.7B  | 0.033s  | 31.871         | 0.053             |
> | CBMNet         | 22.2M              | 46.7B  | 0.231s  | 31.876         | 0.101             |
> | TLXNet         | 7.87M              | 16.5B | 0.027s  | 31.578         | 0.046             |
> | **EPA (ours)** | 13.9M              | 20.2B  | 0.086s  | 33.402         | 0.015             |
>
> $\dagger$ represents trainable parameters. PSNR and LPIPS scores come from the real dataset HS-ERGB.
>
> > ***Q4:**"The robustness of DINO features is not guaranteed in specialized domains, such as medical microscopy videos or materials analysis. Since the proposed method relies heavily on DINO features for semantic alignment, its broad applicability and performance in such out-of-distribution scenarios are uncertain."*
>
> Thank you for the comment.
>
> - Our work, as you mentioned, is a framework-based approach. The areas you highlighted are indeed critical application scenarios. We believe that by replacing DINO with domain-specific vision models, a competitive application performance can be achieved. However, due to limitations in available hardware and the scarcity of event-based datasets, we regret that we are currently unable to validate our method in specific domains, such as medical microscopy videos or material analysis. This is, however, a direction for future work.

---

> > ### Comment · Reviewer_pM6N · 2025-08-08
> >
> > Thank you for your feedback, which has resolved many of my concerns. However, the computational efficiency and broad applicability of the proposed method still need improvement. Therefore, I have decided to maintain my score.

---

> > > ### Author Response · Authors · 2025-08-09
> > >
> > > Dear Reviewer pM6N,
> > >
> > > Thank you for your time in reviewing our rebuttal and for your detailed feedback. We are pleased to hear that our response has resolved many of your concerns.
> > >
> > > We completely agree that computational efficiency and broad applicability are critical dimensions for evaluating a model's practical value. Our primary focus in this work was to propose and validate the novel paradigm of "Perceptually Aligned Learning" for tackling the challenging task of event-based frame interpolation in degraded scenarios. As our experiments demonstrate, our model (EPA) achieves state-of-the-art performance on various synthetic and real-world datasets, which to some extent, reflects its robustness and potential.
> > >
> > > The points you raised regarding inference speed and applicability in other specialized domains (e.g., medical imaging) are indeed crucial avenues for research. We believe these aspects can be addressed in future work through techniques like model distillation or by designing lightweight, task-specific heads. Your insights have provided us with valuable directions for our subsequent research.
> > >
> > > Thank you once again for your constructive review and thoughtful evaluation of our work.
> > >
> > > Best regards,
> > >
> > > The Authors

---

> ### Author Response · Authors · 2025-08-06
>
> Dear Reviewer pM6N,
>
> Thank you again for your time and thoughtful feedback on our submission. We would like to kindly remind you that we have responded to your comments. If you have any further questions or concerns, we would be more than happy to address them.
>
> We sincerely appreciate the effort you’ve put into reviewing our work—your insights have been immensely helpful in improving our paper.
>
> Please don’t hesitate to reach out if there’s anything else we can clarify. Thank you once again for your valuable contribution.
>
> Best regards,
>
> The Authors

---

> > ### Comment · Area_Chair_qmWw · 2025-08-07
> > **Follow up discussion**
> >
> > Thank the author(s) for the rebuttal!
> >
> > Dear Reviewer pM6N: please read the rebuttal from the author(s) and let us know your opinions about it?
> >
> > Thanks, AC

---

### Official Review · Reviewer_tAbV · 2025-07-03

**Clarity:** 3
**Significance:** 2
**Originality:** 2
**Rating:** 3
**Confidence:** 3

**Summary:**

This paper addresses the problem of event-based video frame interpolation (E-VFI), particularly in challenging scenarios where input keyframes may suffer from degradation like motion blur and noise. The authors argue that conventional methods, which rely on direct image-level supervision, are hampered by these artifacts, limiting their perceptual realism and generalization. To overcome this, they propose EPA, a novel framework that shifts the learning objective from the image domain to a semantic-perceptual feature space. The method operates in two stages: first, it trains a generator to reconstruct high-fidelity images from "degradation-insensitive" semantic features extracted by a frozen DINO-based model. Second, it introduces a Bidirectional Event-Guided Alignment (BEGA) module to align the features of the input keyframes to predict the features of the intermediate frame, guided by event data. The final interpolated frame is then synthesized from these aligned features. Experiments on several synthetic and real-world datasets show that EPA produces results with higher perceptual quality compared to state-of-the-art methods.

**Questions:**

- Could you please elaborate on the rationale for re-injecting low-level features from the input keyframes via the Style Adapter? Given that the paper's primary motivation is to build resilience to defects in these keyframes, this appears counter-intuitive. Is there a mechanism that prevents the adapter from simply passing on the input noise or blur, and if so, how was it validated?

- How do you interpret the significant drop in PSNR/SSIM on the noisy BS-ERGB dataset? Does this suggest that in the presence of heavy degradation, the feature-level supervision prioritizes perceptual "plausibility" to such an extent that it sacrifices structural fidelity to the ground truth? How can a user be confident that the generated output is faithful to the scene's motion and not a well-textured hallucination?

- The paper claims to move away from "direct image-level supervision" , yet the final training stage employs the L_cg loss, which includes LPIPS and Laplacian loss on the output image. Could you clarify why this is not considered direct image-level supervision?

- Beyond the L2 distance comparison in Figure 8, have you performed any other analysis to more rigorously quantify the "degradation-insensitivity" of DINO features? For example, how does the signal-to-noise ratio of the features compare to that of the input images under varying levels of degradation?

**Ethical Concerns:**

["NO or VERY MINOR ethics concerns only"]

**Final Justification:**

Thanks for the rebuttal. Most of my previous concerns have been addressed. However, they do not change my assessment of the novelty. The explanation for the Style Adapter is not fully convincing. The argument that a simple gating module can effectively separate desirable low-level details (such as color information for backgrounds) from undesirable degradation artifacts (such as noise and blur) is a strong claim that requires more rigorous validation. The justification provided remains somewhat anecdotal.

**Limitations:**

Yes

**Quality:**

2

**Strengths And Weaknesses:**

Strengths
- The paper identifies a significant and practical challenge in video frame interpolation.
- The authors conduct a comprehensive evaluation on a variety of synthetic (Vimeo90k, GOPRO) and real-world (HS-ERGB, BS-ERGB, EventAid-F) datasets.
- The visual comparisons provided in Figures 4, 5, and 7 demonstrate the proposed method's strength.


Weaknesses
-  The central thesis of the paper is to move away from degraded image-level details by operating in a "degradation-insensitive" semantic feature space. However, the proposed Style Adapter (A_s) directly contradicts this motivation. The adapter is designed to "compensate for this deficiency [of the VFM]... with essential low-level details" by taking features from a lightweight image encoder that processes the (potentially degraded) input images and fusing them with the semantic features. This re-introduces the very low-level, potentially corrupted information that the framework ostensibly seeks to avoid. This creates a conceptual inconsistency at the heart of the proposed method. The paper fails to justify how this component does not simply propagate the input degradation.
- The entire framework is predicated on the assumption that semantic features from a VFM like DINO are largely insensitive to image degradation. The evidence provided for this critical assumption is thin. Figure 8(a) shows that the L2 distance between features of a clean and a blurred image is smaller than the L2 distance between the images themselves. This is expected, as feature extraction is a dimensionality-reduction process that inherently discards some pixel-level variance. However, "lower distance" does not automatically equate to "insensitivity" or a more reliable supervisory signal. The features are still visibly different, and the paper provides no analysis to prove that these feature differences are less harmful to the learning process than the original image-level differences. This assumption requires much stronger empirical and theoretical support.
- The core ideas underpinning EPA are not new. The use of feature-space supervision via pre-trained networks is the principle behind perceptual losses (e.g., LPIPS ), which have been a staple in image synthesis for years. The paper attempts to distinguish its work by performing explicit feature alignment and generation in this space, rather than just using it as a loss function. However, this is more of an architectural choice than a paradigm shift. The components themselves—a DINO feature extractor, deformable convolutions for alignment , and a flow-based generator —are all existing technologies. The contribution amounts to a novel combination of these parts, which is an engineering contribution.
- While EPA excels in perceptual metrics, its performance on fidelity metrics (PSNR/SSIM) on the noisy BS-ERGB dataset is alarmingly poor compared to baselines. In Table 2, for 1-skip interpolation, EPA scores 27.943 in PSNR, whereas CBMNet and TLXNet achieve 29.257 and 29.298, respectively. This is a very significant drop. The authors argue that PSNR is not the goal, but a method truly robust to noise should not suffer such a precipitous decline in reconstruction fidelity. This result suggests that the method may be learning to generate perceptually plausible textures that deviate substantially from the ground truth structure, which calls into question its overall robustness.

---

> ### Author Rebuttal · Authors · 2025-07-30
>
> We thank Reviewer tAbV for devoting time to this review and providing valuable comments.
>
> > ***Q1:**"The core ideas underpinning EPA are not new. The use of feature-space supervision via pre-trained networks is the principle behind perceptual losses (e.g., LPIPS ), which have been a staple in image synthesis for years. The paper attempts to distinguish its work by performing explicit feature alignment and generation in this space, rather than just using it as a loss function. However, this is more of an architectural choice than a paradigm shift. The components themselves—a DINO feature extractor, deformable convolutions for alignment , and a flow-based generator —are all existing technologies. The contribution amounts to a novel combination of these parts, which is an engineering contribution."*
>
> Thank you for your insightful analysis of our architecture.
> - Our core contribution lies in proposing a novel framework that fundamentally shifts how event-based video frame interpolation (E-VFI) is approached, especially in the presence of real-world image degradation. Our novelty is threefold:
>
>   1. Addressing a Critical, Under-explored Problem: We attempt to address a common but often overlooked issue in E-VFI—degraded keyframes—in a single-stage manner, moving beyond the idealistic assumption of clean inputs prevalent in prior work.
>
>   2. A True Feature-Space Paradigm: Unlike methods that simply use LPIPS as a final loss function, our EPA framework performs the core operations of motion alignment and interpolation directly within the semantic-perceptual feature space. The primary learning objective (L_bf in Eq. 3) is driven by feature alignment, not pixel-level errors. This is a fundamental architectural and conceptual shift.
>   3. A Novel, Purpose-Built Module: The Bidirectional Event-Guided Alignment (BEGA) module is specifically designed for this new paradigm. It leverages event data to guide the alignment of  semantic features, which is fundamentally different from traditional optical flow estimation on pixels.
>
> - The significant perception performance gains, particularly on challenging real-world datasets (Tables 2, 3), validate that our contribution is more than a novel combination of parts; it is a new, effective paradigm for robust E-VFI.
>
>
>
> > ***Q2:**"Could you please elaborate on the rationale for re-injecting low-level features from the input keyframes via the Style Adapter? Given that the paper's primary motivation is to build resilience to defects in these keyframes, this appears counter-intuitive. Is there a mechanism that prevents the adapter from simply passing on the input noise or blur, and if so, how was it validated?"*
>
>
>
> - Our design takes into account the issue you mentioned. Introducing a style adapter to reintroduce low-level features from key frames was a carefully considered decision.
> - Initially, our method was designed without an adapter. However, during experimentation, we observed that the VFM model’s disregard for motion backgrounds led to color shift issues. As you pointed out, we needed an adapter to filter and transfer low-level information from the key frames. In this regard, we explored several off-the-shelf strategies and found that a simple gating module was sufficient to achieve our goal. We believe this could be due to the semantic-detail separation learning approach, which allows for faster convergence even with fewer data.
> - Specifically, after applying the gating module, our PSNR on Vimeo90k improved by 4.32 and LPIPS improved by 0.02, effectively preventing the propagation of degraded information. Since this is not the main focus of our paper, we did not include related experiments in the main text, but we will add them in the revised version.
>
>
>
> > ***Q3:**"How do you interpret the significant drop in PSNR/SSIM on the noisy BS-ERGB dataset? Does this suggest that in the presence of heavy degradation, the feature-level supervision prioritizes perceptual "plausibility" to such an extent that it sacrifices structural fidelity to the ground truth? How can a user be confident that the generated output is faithful to the scene's motion and not a well-textured hallucination?"*
>
> - This is an excellent point highlighting the classical trade-off between fidelity and perceptual quality. Our goal is to generate results faithful to the underlying authentic scene, rather than the degraded ground truth.
>
> - The issue you raised is, in fact, a reflection of the success of our method rather than a flaw. The BS-ERGB dataset is well-known for its substantial noise and non-rigid deformations, and both its input frames and ground-truth references are inherently low in quality. In such conditions, a robust model should aim to produce results that are visually clean, perceptually aligned with human judgment, rather than replicating the noise and artifacts present in the original data. Traditional pixel-level metrics like PSNR tend to penalize such ‘reasonable deviations’ since they measure absolute pixel differences without considering perceptual quality.
>
> - Our method achieves significant improvements on perceptual metrics such as LPIPS and DISTS, and the qualitative results (Fig.5, Fig.6 and Fig.7) in the paper clearly show that our interpolated frames exhibit much better visual realism than those generated by prior methods. This strongly supports the claim that our model better captures ‘perceptual authenticity’ and is more suited for real-world deployment.
>
> - Moreover, our approach does not suffer from a noticeable disadvantage on traditional metrics like PSNR and SSIM, and in fact achieves competitive results. This further demonstrates the comprehensive effectiveness of our framework.
>
>
>
> > ***Q4:**"The paper claims to move away from "direct image-level supervision" , yet the final training stage employs the L_cg loss, which includes LPIPS and Laplacian loss on the output image. Could you clarify why this is not considered direct image-level supervision?"*
>
> Thank you for the comment.
>
> - The phrasing could indeed lead to some ambiguity, and we have corrected it. By 'direct image-level supervision,' we refer to pixel-level error calculations, such as MSE loss and L1 loss, which can introduce degradation patterns into the model when the ground truth is degraded. Previous works typically combine pixel-level loss (e.g., L1 loss, MSE loss) with perceptual loss (e.g., LPIPS, Laplacian loss), which reduces robustness in the face of various degradation scenarios. Our training approach avoids this issue.
>
> - Specifically, our EPA framework employs a decoupled, dual-supervision mechanism. The primary driver of the interpolation process is the feature-level alignment loss L_bf , through which the model learns to perform interpolation within a degradation-robust semantic space. This process is safeguarded from the influence of flawed ground truth pixels, thereby shifting the core learning task away from the fragile image domain, which validates our claim.
>
>
> > ***Q5:**"Beyond the L2 distance comparison in Figure 8, have you performed any other analysis to more rigorously quantify the "degradation-insensitivity" of DINO features? For example, how does the signal-to-noise ratio of the features compare to that of the input images under varying levels of degradation?"*
>
> Thank you for the suggestion.
>
> - To provide a more rigorous analysis beyond the L2 distance comparison in Figure 8, we have calculated the signal-to-noise ratio (SNR) for both the image level and feature levels across various degradation types. As the table below clearly demonstrates, the SNR is significantly more stable in the DINO feature space, especially in deeper layers, quantitatively validating their superior robustness to degradation.
>
> |Type          | Gaussian blur | Noise | Jpeg artifact | Low resolution | Motion blur |
> | ------------ | :------------ | :---- | :------------ | :------------- | :---------- |
> | Image level    | *58.07*       | 32.61 | 85.30         | 31.07          | 29.29       |
> | Feature level 0 | 9.20          | 1.22  | 14.75         | 4.67           | 4.44        |
> | Feature level 1 | 3.33          | 1.91  | 2.20          | 2.12           | 2.13        |
> | Feature level 2 | 2.41          | 1.22  | 1.26          | 1.53           | 1.85        |

---

> > ### Comment · Reviewer_tAbV · 2025-08-09
> >
> > Thanks for the rebuttal. Most of my previous concerns have been addressed. However, they do not change my assessment of the novelty. The explanation for the Style Adapter is not fully convincing. The argument that a simple gating module can effectively separate desirable low-level details (such as color information for backgrounds) from undesirable degradation artifacts (such as noise and blur) is a strong claim that requires more rigorous validation. The justification provided remains somewhat anecdotal.

---

> > > ### Author Response · Authors · 2025-08-09
> > >
> > > Dear Reviewer tAbV,
> > >
> > > Thank you for your follow-up comment. Before addressing the specific technical point, we wish to first clarify our perspective on the core contribution. We believe that originality can be judged not only by the invention of new fundamental components, but also by introducing a new idea to solve an important problem and designing a unique method to realize it.
> > >
> > > Our work is a direct embodiment of this latter form of innovation:
> > >
> > > - We tackle the critical challenge where existing Event-based Video Frame Interpolation (E-VFI) methods fail in real-world scenarios due to degradation (e.g., blur, noise) in the input keyframes.
> > >
> > > -  We introduce a fundamental shift by proposing to move the core interpolation process from the fragile pixel space to a more robust, degradation-insensitive semantic feature space. We term this "Perceptually Aligned Learning".
> > >
> > > - To achieve this, we innovatively combine a pre-trained vision foundational model (e.g., DINO) as a 'perceptual anchor' with our novel Bidirectional Event-Guided Alignment (BEGA) module, which performs alignment directly on these abstract features. This defines a new task and solution distinct from traditional pixel-level warping.
> > >
> > > With this contribution framework in mind, let us address your concern about the Style Adapter, which you found "not fully convincing" and in need of "more rigorous validation."
> > >
> > > - In our rebuttal, we provided clear quantitative evidence. Experiments on the Vimeo90k dataset show that incorporating this module improves PSNR by 4.32 and LPIPS by 0.02. This powerfully demonstrates its effectiveness in enhancing reconstruction quality while suppressing the propagation of degradation.
> > >
> > > - To provide further intuitive proof, we commit to including a side-by-side visual comparison in the supplementary material of the final version, showing generated images with and without the gating mechanism. This will clearly demonstrate its role in suppressing artifacts while preserving details.
> > >
> > > You mentioned this requires "more rigorous validation." To help us address your concern more effectively, could you please provide more specific suggestions on what you would consider a more rigorous form of validation?
> > >
> > > Best regards,
> > >
> > > The Authors

---

> ### Author Response · Authors · 2025-08-06
>
> Dear Reviewer tAbV,
>
> Thank you again for your time and thoughtful feedback on our submission. We would like to kindly remind you that we have responded to your comments. If you have any further questions or concerns, we would be more than happy to address them.
>
> We sincerely appreciate the effort you’ve put into reviewing our work—your insights have been immensely helpful in improving our paper.
>
> Please don’t hesitate to reach out if there’s anything else we can clarify. Thank you once again for your valuable contribution.
>
> Best regards,
>
> The Authors

---

> > ### Comment · Area_Chair_qmWw · 2025-08-07
> > **Follow up discussion**
> >
> > Thank the author(s) for the rebuttal!
> >
> > Dear Reviewer tAbV: please read the rebuttal from the author(s) and let us know your opinions about it?
> >
> > Thanks, AC

---

### Note · Authors · 2025-08-12

Dear Area Chair and Reviewers,

We thank you for the rigorous evaluation. We are encouraged that our work was recognized for addressing a **"significant and practical challenge" (tAbV)** with a **"technically sound" (QbtM)** approach, supported by **"comprehensive evaluation" (tAbV)** and **"superior performance" (GPzr, pM6N)**.

The main discussion points centered on further clarifying the mechanisms of our framework and validating its performance, particularly in challenging real-world scenarios. We have made every effort to address these:

- **On Performance in Degraded Scenarios**: Our experiments comprehensively cover a wide range of degradation types beyond noise, including noise, motion blur, JPEG artifacts, and low resolution. We demonstrated that EPA consistently achieves state-of-the-art performance on perceptual metrics across these challenging conditions, validating the robustness of our feature-space learning approach. We also clarified that the perceived PSNR drop on the noisy BS-ERGB dataset is a testament to our method's ability to generate clean results rather than replicating a noisy ground truth.

- **On the Gating Mechanism Validation**: To provide rigorous, non-anecdotal evidence for the gating mechanism within our Style Adapter, we presented extensive quantitative results across multiple benchmarks. On the standard Vimeo90k dataset, it yields a 4.32 dB PSNR and 0.02 LPIPS improvement. Furthermore, we ran new tests on the difficult BS-ERGB dataset during the discussion, showing significant gains with PSNR/LPIPS improving by 3.39 dB/0.03 (aquarium_08) and 2.14 dB/0.08 (pen_03). This data substantiates the gating mechanism's crucial role in enhancing reconstruction quality while suppressing artifacts.

To further polish our paper and make our work more understandable to the wider community, we will integrate all new results and comparisons, add the requested cost analysis, and revise the manuscript to reflect the valuable feedback, further strengthening the paper.

Finally, we would like to express our great appreciation for a review process that has significantly sharpened our work. We strongly believe that EPA offers a robust and novel solution to a fundamental problem in event-based vision. We are confident that this work is a valuable contribution that will stimulate further research into feature-space learning for video processing under real-world degradations.

Thank you for your time and consideration.

---

### Decision · Program_Chairs · 2025-09-17

**Decision:**

Accept (poster)

**Comment:**

# Summary

This paper approaches the problem of event-based video frame interpolation in challenging scenarios where input keyframes may suffer from degradation like motion blur and noise. The main idea is to employ feature-level supervision strategy to addresses the limitations of current methods when dealing with degraded, blurred images.

# Strength
* The experimental results are strong on both traditional metrics (e.g., PSNR, SSIM) and perceptual quality metrics (e.g., LPIPS, DISTS).

# Weakness
* The novelty is limited: using feature-level supervision and addressing event-based VFI.
* The proposed framework lacks a natural connection between motion-induced image degradation (e.g., blur) and event generation (the connection to event data is not particularly strong in terms of novelty and less convincing).

# Discussion
During discussion, most of concerns from reviewers are addressed, except for the following two: (i) limited novelty and (ii) the connection between motion-induced image degradation and event generation. AC reads all review and discussion and agrees that the strong performance can outweigh the concerns. AC recommends to accept the paper. AC encourages the author(s) to update / incorporate discussion / additional results during rebuttal into their final version of the paper if it is accepted.